# The Ethanol Extracts of *Osmanthus fragrans* Leaves Ameliorate the Bone Loss via the Inhibition of Osteoclastogenesis in Osteoporosis

**DOI:** 10.3390/plants12020253

**Published:** 2023-01-05

**Authors:** Yo-Seob Seo, HyangI Lim, Jeong-Yeon Seo, Kyeong-Rok Kang, Do Kyung Kim, Hyun-Hwa Lee, Deuk-Sil Oh, Jae-Sung Kim

**Affiliations:** 1Institute of Dental Science, Chosun University, Gwangju 61452, Republic of Korea; 2Department of Oral and Maxillofacial Radiology, School of Dentistry, Chosun University, Gwangju 61452, Republic of Korea; 3Department of Biology, Chosun University, Gwangju 61452, Republic of Korea; 4Jeollanamdo Forest Resources Research Institute, Naju 58213, Republic of Korea

**Keywords:** osteoporosis, osteoclast, receptor activator of nucleus factor-κB ligand, osteoclastogenesis, anti-inflammation, *Osmanthus fragrans*

## Abstract

The aim of this study was to evaluate the anti-osteoporosis effects of *Osmanthus fragrans* leaf ethanol extract (OFLEE) in bone marrow-derived macrophages (BMM) and animals with osteoporosis. OFLEE not only suppressed tartrate-resistant acid phosphatase (TRAP)-positive cells with multiple nuclei but also decreased TRAP activity in BMM treated with macrophage colony stimulating factor (M-CSF) and receptor activator of nuclear factor-κB (NF-κB) ligand (RANKL). The formation of F-actin rings and the expression and activation of matrix metalloproteinases were decreased by OFLEE in BMM treated with M-CSF and RANKL. OFLEE suppressed M-CSF- and RANKL-induced osteoclastogenesis by inhibiting NF-κB phosphorylation, tumor necrosis factor receptor-associated factor 6, c-fos, the nuclear factor of activated T-cells, cytoplasmic 1, and cathepsin K in BMM. OFLEE downregulated reactive oxygen species, cyclooxygenase-2, inducible nitric oxide synthase, prostaglandin E_2_, tumor necrosis factor α, interleukin (IL)-1β, IL-6, IL-17, and RANKL in BMM treated with M-CSF and RANKL. Oral administration of OFLEE suppressed osteoporotic bone loss without hepatotoxicity in ovariectomy-induced osteoporosis animals. Our findings suggest that OFLEE, with anti-inflammatory effects, prevents osteoporotic bone loss through the suppression of osteoclastic differentiation in BMM and animals with osteoporosis.

## 1. Introduction

Osteoporosis (OP) is a chronic skeletal disorder with the increased risk of bone fracture leading to microarchitectural deterioration, caused by the decrease of bone mass related to skeletal fragility [1,2]. The prevalence of OP is estimated higher by approximately 33% in women, more than 20% of men [3]. As the global elderly population increases, the rapidly increasing prevalence of OP is being considered as a public health problem accompanied with the social and economic burden worldwide [1].

Generally, the homeostasis of a normal bone is precisely balanced between the bone resorption by osteoclast and the bone formation by osteoblast [1]. Osteoclasts, multinucleated giant cells differentiated from the cell-to-cell fusion of monocytes and macrophage by macrophage colony stimulating factor (M-CSF) and receptor activator of nucleus factor-κB ligand (RANKL) [4], can induce bone resorption through acidification and proteolytic digestion [1], whereas osteoblasts originated from pluripotent mesenchymal stem cells and are involved with the bone formation through the synthesis of the bone matrix component, including type I collagen and the bone matrix mineralization caused by the deposition of calcium phosphate crystals such as hydroxyapatite [5]. Hence, the imbalance of bone homeostasis between osteoclast-induced bone resorption and osteoblast-induced bone formation is closely associated with the pathogenesis of bone diseases. Especially, OP is caused by the excess of osteoclast-induced bone resorption, more than the osteoblast-induced bone formation at bone resorptive cavity [6]. The increase of inflammation caused by pathophysiological risk factors such as the alteration of sexual hormones, aging, obesity, low calcium and the deficiency of vitamin D is closely associated with the acceleration of bone resorption though the induction of bone homeostasis imbalance between osteoclast and osteoblast [3,7]. Hence, the suppression of osteoclastogenesis through the down-regulation of inflammation is to be a preventive strategy for OP. 

*Osmanthus fragrans* (*O. fragrans*) is a medicinal and edible plant used traditionally for toothache, rheumatoid arthritis, asthma, coughs, physical pain and menopausal pain in Korea, Japan and southwestern China [8,9]. Especially, the flower of *O. fragrans*, with its fragrance, is not only used as a cosmetic substance, but also is used traditionally as a food additive for tea and beverages [10]. In addition, *O. fragrans* leaves (OFL) are also used as a traditional folk medicine to relive pain and coughs [10]. Recent pharmacological studies reported that *O. fragrans*, with more than 183 compounds such as flavonoids and polyphenols, have various biological activities including antioxidant and anti-inflammation properties [8]. Recent studies reported on the anti-inflammatory effect of OFL containing a phenylpropyl triterpenoids and phillyrin in the macrophages treated with lipopolysaccharide [9,11]. Furthermore, Zhang et al., reported that the phillyrin prevented osteolysis via the inhibition of osteoclast differentiation [12]. Hence, we hypothesized that the OFL, with its anti-inflammatory effects, might suppress or delay the osteoclastic bone loss through the downregulation of pro-inflammatory factors associated with the osteoclastogenesis of macrophages. 

Based on our hypothesis, the aim of the present study was to evaluate the anti-osteoporosis effects of the OFL ethanol extracts (OFLEE) in primary bone marrow derived macrophage (BMM) and experimental animals with OP.

## 2. Results

### 2.1. OFLEE Does Not Induce Cell Death in Either L929 Fibroblast Cells or BMMs

To determine whether the OFLEE increases the cytotoxicity in BMMs, MTT assays were performed in the mouse embryonic fibroblast L929 cells used as normal cells and BMMs treated with 1, 10, 25, 50, and 100 μg/mL OFLEE for 24 h. As shown in Figure 1A, 1–100 μg/mL OFLEE did not increase the cytotoxicity in L929 fibroblast cells. Although 1 μg/mL OFLEE did not increase the viability of BMMs, viabilities of BMMs were increased by 10 (*p* < 0.05), 25 (*p* < 0.01), 50 (*p* < 0.01) and 100 μg/mL OFLEE (*p* < 0.01) for 24 h. Hence, these data indicated that the 1–100 μg/mL OFLEE did not induce cell death in either L929 fibroblast cells or BMMs.

### 2.2. OFLEE Suppresses the Osteoclastic Differentiation of BMMs in the Presence of M-CSF and RANKL

BMMs were stimulated with 10, 25, and 50 μg/mL OFLEE in the presence of 30 ng/mL M-CSF and 50 ng/mL RANKL for 72 h. Thereafter, to investigate the alteration of osteoclastic differentiation, tartrate-resistant acid phosphatase (TRAP) staining using a TRAP and alkaline phosphatase (ALP) double-stain kit (TaKaRa Bio Inc., Kyoto, Japan) was performed, as shown in Figure 2A,B. TRAP-positive cells with multiple nuclei were significantly increased in the BMMs treated with M-CSF and RANKL, whereas TRAP-positive cells were not observed in untreated control BMMs. However, TRAP-positive cells treated with M-CSF and RANKL were decreased by OFLEE in a dose-dependent manner. Moreover, the activity of TRAP was significantly decreased by OFLEE in the BMMs treated with M-CSF and RANKL, as shown in Figure 2C. Hence, these data indicated that the OFLEE suppressed the osteoclastic differentiation of BMMs treated with M-CSF and RANKL, in a dose-dependent manner.

### 2.3. OFLEE Inhibits the Expression and Activation of Matrix Metalloproteinase via the Suppression of F-Actin Formation in the BMMs Treated with M-CSF and RANKL

To investigate the formation of F-actin ring, which is a prerequisite for osteoclast bone resorption [13], BMMs were stimulated with 10, 25, and 50 μg/mL OFLEE in the presence of 30 ng/mL M-CSF and 50 ng/mL RANKL for 72 h. Thereafter, F-actin staining was performed as shown in Figure 3A. The formation of the F-actin ring was significantly increased in the BMMs treated with M-CSF and RANKL, whereas the F-actin ring was not observed in untreated control BMMs. However, the formation of the F-actin ring was decreased by OFLEE in the BMMs treated with M-CSF and RANKL. Moreover, as shown in Figure 3B,C, OFFLEE not only suppressed the expression of matrix metalloproteinases (MMP) associated with the activation of osteoclast such as MMP-2 and MMP-9 [14], but also decreased the activation of MMPs in the BMMs treated with M-CSF and RANKL. Hence, these findings suggested that OFLEE suppressed the activation of osteoclasts through the inhibition of F-actin formation and the suppression of MMPs expression and activation during the M-CSF and RANKL-induced osteoclastic differentiation of BMMs.

### 2.4. OFLEE Suppresses the Osteoclastic Differentiation of BMMs through the Inhibition of Nucleus Factor-κB (NF-κB) Phosphorylation in the Presence of M-CSF and RANKL

RANKL-induced osteoclastogenesis is generally initiated via the activation of NF-κB in monocytes and macrophages [15]. Hence, to investigate the alteration of NF-κB phosphorylation, BMMs were stimulated with 10, 25, and 50 μg/mL OFLEE in the presence of 30 ng/mL M-CSF and 50 ng/mL RANKL for 72 h. As shown in Figure 4A, OFLEE effectively suppressed the RANKL-induced NF-κB phosphorylation in BMMs. Furthermore, OFLEE downregulated the expression of transcriptional factors associated with osteoclastogenesis, such as tumor necrosis factor receptor-associated factor 6 (TRAF6), c-fos, nuclear factor of activated T-cells, cytoplasmic 1 (NFATc1), and cathepsin K in the BMMs treated with M-CSF and RANKL (Figure 4B). Hence, these findings clearly indicated that OFLEE suppressed the M-CSF and RANKL-induced osteoclastogenesis through the suppression of TRAF6-NF-κB-NFATc1 axis in BMMs.

### 2.5. OFLEE-Mediated Anti-Osteoclastogenesis Is Mediated by the Suppression of Reactive Oxygen Species (ROS), Inflammatory Mediators, Pro-Inflammatory Cytokines, RANKL, and RANK in the BMMs Treated with M-CSF and RANKL

Inflammation is closely associated with the osteoclastogenesis of monocytes and macrophages [16]. Hence, to investigate the OFLEE-mediated-anti-inflammatory effects, BMMs were stimulated with 10, 25, and 50 μg/mL OFLEE in the presence of 30 ng/mL M-CSF and 50 ng/mL RANKL for 72 h. As shown in Figure 5A, OFLEE significantly suppressed the production of ROS production in the BMMs treated with M-CSF and RANKL. Furthermore, the expression of inflammatory mediators such as inducible nitric oxide synthase (iNOS), cyclooxygenase-2 (COX-2) and prostaglandin E_2_ (PGE_2_) was also decreased by OFLEE in the BMMs treated with M-CSF and RANKL (Figure 5B,C). Moreover, the expression of pro-inflammatory cytokines associated with the osteoclastic differentiation such as tumor necrosis factor α (TNFα), interleukin (IL)-1β, IL-6, and IL-17 were down-regulated by OFLEE in the BMMs treated with M-CSF and RANKL, as shown in Figure 6A. Recently, RANKL has been considered as the final downstream cytokine to induce bone resorption through the regulation of osteoclastogenesis [17]. Hence, to investigate whether OFLEE is involved with the expression of RANKL, BMMs were stimulated with 10, 25, and 50 μg/mL OFLEE in the presence of 30 ng/mL M-CSF and 50 ng/mL RANKL for 72 h. As shown in Figure 6B, RANKL and M-CSF not only increased the expression of RANKL, but also up-regulated the expression of its receptor, RANK, in BMMs, whereas OFLEE counteracted significantly the RNAKL and M-CSF-induced RANKL and RANK expression in BMMs. Hence, these findings suggested that OFLEE suppressed the M-CSF and RANKL-induced osteoclastic differentiation through the down-regulation of ROS, inflammatory mediators, pro-inflammatory cytokines, RANKL, and RANK in BMMs.

### 2.6. The Oral Administration of OFLEE Suppresses Osteoporotic Bone Loss without Hepatotoxicity in the Animals with Osteoporosis Generated by Ovariectomy

To verify the OFLEE-induced anti-osteoclastogenesis in vivo, experimental animals with osteoporosis were generated by surgical ovariectomy (OVX). Thereafter, 5 and 10 mg/kg OFLEE per body weight were orally supplied once daily for 9 weeks to experimental animals with osteoporosis. As shown in Figure 7A, there were no deaths and the alteration of body weight caused by the supplementation of vehicle or OFLEE in the animal groups was insignificant during the experiment periods. In addition, as shown in Figure 7B, the concentration of alanine aminotransferase (ALT) and aspartate aminotransferase (AST) in the sera collected from experimental animals showed no statistical significance between all of the animal groups. Hence, these data indicated that OFLEE had no hepatotoxicity or hepatic injury in experimental animals during the experimental periods. The results of micro computed Tomography (μCT) showed that the bone loss was significantly increased in the femoral bone of animals with OVX (*n* = 5), compared with that of the naïve (*n* = 5) and sham (*n* = 5) groups, whereas the oral supplementation of 5 mg/kg (*n* = 5) or 10 mg/kg (*n* = 5) OFLEE significantly suppressed the bone loss in the femoral bone of animals performed with OVX, as shown in Figure 7C. Moreover, the results of trabecular morphometric parameters included bone mineral density (BMD), bone volume per tissue volume (BV/TV), trabecular number (Tb.N) and trabecular thickness (Tb.Th), which were significantly decreased in the OVX group compared with naive, shame, OVX orally supplemented with 5 mg/kg or 10 mg/kg OFLEE groups, whereas trabecular separation (Tb.Sp) was increased in the OVX groups, compared with naïve, sham, OVX orally supplemented with 5 mg/kg or 10 mg/kg OFLEE groups, as shown in Figure 7D. Furthermore, hematoxylin and eosin (H&E) staining showed the increase of empty lacunas and osteoclasts (indicated by a black arrow) in the femoral bones dissected from the OVX group compared with naïve, sham, OVX orally supplemented with 5 mg/kg or 10 mg/kg OFLEE groups, as shown in the upper panel of Figure 8. Moreover, Masson trichrome staining showed that new bone collagen fibers, stained a blue color, and mature bone, stained a red color, were higher than naïve, sham, OVX orally supplemented with 5 mg/kg or 10 mg/kg OFLEE groups compared with OVX group, as shown in the lower panel of Figure 8. Hence, these results indicated that the oral administration of OFLEE suppressed the osteoporotic bone loss without hepatotoxicity in the animals with osteoporosis.

## 3. Discussion

Excessive bone loss caused by increased osteoclast activity acts as a pathogenesis of OP [16]. Osteoclasts associated with bone resorption are multinucleated giant cells caused by cell-to-cell fusion through the osteoclastic differentiation of monocyte and macrophages regulated by M-CSF and RANKL [16]. M-CSF is related to the survival and motility of RANKL-induced osteoclast [16]. However, the bonding of RANKL to its receptor RANK induces the phosphorylation of NF-κB and mitogen-activated protein kinases (MAPK) via the expression of TRAF6 [18]. Sequentially, the activation of NFATc1, a transcriptional factor, induces the expression of osteoclast specific genes such as TRAP, RANKL and c-fos in osteoclast precursors [18]. In addition, the maturation of osteoclasts is needed for the formation of podosomes, which are unique actin structures accumulated in a ring-like structure called an F-actin ring at terminal osteoclastogenesis [19]. Therefore, the formation of actin ring is essential for the bone resorption [20]. In the present study, we demonstrated that the OFLEE suppressed the TRAP-positive cells with multiple nuclei, and that the formation of F-actin ring and the expression of Cathepsin K and MMPs are associated with bone resorption through the down-regulation of transcriptional factors such as TRAF6, NF-κB, c-fos, and NFATc1 in the BMM treated with M-CSF and RANKL. Hence, our findings suggest the OFLEE suppresses the osteoclastic bone loss through the inhibition of M-CSF and RANKL-induced osteoclastogenesis in osteoclast progenitors.

However, osteoporosis is closely associated with inflammation [21]. Many studies have reported that excessive ROS production is closely associated with RANKL-induced osteoclastogenesis through the activation of the TRAF6-NF-κB axis or TRAF6-MAPKs axis [22,23,24]. Hence, many studies suggest that the down-regulation of ROS-TRAF6 axis mediated osteoclastogenesis is capable of preventing OP [25,26]. In addition, inflammatory mediators such as iNOS, COX-2 and PGE_2_ in osteoclast precursors are involved with RANKL-induced osteoclastogenesis. Kondo et al. reported that the iNOS promoted osteoclastogenesis under hypoxic culture condition [27]. Cuzzocrea et al. reported that the inhibition of iNOS not only prevented OVX-induced bone loss, but also suppressed the upregulation of pro-inflammatory cytokines such as IL-1β, IL-6 and TNFα [28]. Furthermore, Han et al. reported that RANKL selectively induced COX-2 and its downstream inflammatory mediator PGE_2_ via the expression of Rac family small GTPase 1 (Rac1) in osteoclast precursors [29]. The celecoxib, a COX-2 inhibitor, inhibited the osteoclastic differentiation of BMM [29]. Thus, these suggest that the suppression of inflammatory mediators is a preventive strategy for OP via the inhibition of osteoclastogenesis. Sequentially, ROS or inflammatory mediator-dependent pro-inflammatory cytokines are closely associated with osteoclastogenesis [30]. Similarly to RANKL, TNFα activates the transcriptional factors such as NF-κB, cFos, and NFATc1 to induce osteoclastic differentiation via the recruitment of TRAFs, including TRAF6, and induces the expression of RANKL [31,32,33]. De Vries et al. reported that the infliximab, a TNFα antagonist, inhibited the formation of osteoclasts in peripheral blood monocytes. IL-1β, a representative pro-inflammatory cytokine, acted as a pathophysiological risk factor in various inflammatory diseases and is involved with osteoclastogenesis via the expression of RANKL by marrow stromal cells and osteoblast [34,35,36]. Kitazawa et al. reported that the IL-1 receptor antagonist (IL-1RA) suppressed the bone resorption via the inhibition of osteoclastogenesis in OVX mice [37]. Many studies have reported that IL-6 is involved with osteoclastogenesis via the upregulation of IL-1β and TNFα [38,39], whereas He et al. reported that the inhibition of IL-6 using its neutralizing antibody not only suppressed the TRAP-positive cells with multiple nuclei, but also alleviated the bone loss in the mice model generated by microgravity [40]. IL-17 is not only a representative cytokine related with osteoclastogenesis in human monocytes, but also induced the expression of RANKL [41,42,43]. Furthermore, Funaki et al. reported that resolvin E1-mediated anti-osteoclastogenesis was mediated by the suppression of IL-17-induced RANKL expression. Taken together, these indicate that the down-regulation of pro-inflammatory cytokines, including TNFα, IL-1β, IL-6, and IL-17, could prevent bone loss through the suppression of RANKL expression in osteoclast precursors. In the present study, we demonstrated that OFLEE significantly suppressed the production of ROS, the expression of inflammatory mediators such as iNOS, COX-2, and PGE_2_, and proinflammatory cytokines associated with osteoclastogenesis such as TNFα, IL-1β, IL-6, IL-17, and RANKL in the BMM treated with M-CSF and RANKL. Thus, our findings suggest that OFLEE-mediated anti-osteoclastogenesis is mediated by the suppression of ROS, inflammatory mediators, and pro-inflammatory cytokines. 

As a clinical intervention for patients with OP, estrogen as a hormonal therapy, calcitonin, bisphosphonates, and teriparaide were administered to increase BMD or to prevent bone loss. However, there are several limitations, such as the increase of cardiovascular events and breast cancer risk, limited efficacy and increase of cancer risk for long-term medication, atypical femur fractures and osteonecrosis of the jaw, and the increase of osteosarcoma risk [44]. Therefore, it is necessary to develop a therapeutic or preventive substance with the effective suppression of bone loss, biological safety, and lower side effects for long-term mediation for patients with OP. In the present study, we demonstrated that the oral supplementation of OFLEE for 9 weeks significantly alleviated bone loss without hepatotoxicity and body weight change in the animals with OP generated by OVX. Thus, our data indicate that OFLEE prepared from OFL, an edible and medicinal plant, not only has biological safety for long-term medication, but also prevents bone loss through the suppression of osteoclastogenesis, resulting in the down-regulation of ROS production, inflammatory mediators, and pro-inflammatory cytokines in animals with OP.

## 4. Materials and Methods

### 4.1. Extraction of OFL

OFL were collected from the garden of the Jeollanamdo Forest Resources Research Institute (JFRRI; Naju, Jeonnam 58213, Republic of Korea) and identified morphologically and genetically by JFRRI. Then, 100 g of collected OFL was soaked in 1000 mL of 80% ethanol (ethanol:water = 80:20, *v/v*) at room temperature for 7 days in a dark environment. Collected 80% ethanol containing crude OFL extract was filtered through Whatman filter paper and concentrated using a rotary evaporator. Thereafter, prepared OFLEE was frozen at –80 °C and dried in a freeze-dryer. Finally, 100 mg of dried OFLEE was dissolved in 1 mL 100% ethanol to be used in this study.

### 4.2. Isolation and Cultivation of BMMs

All animal procedures were approved by the Chosun University Institutional Animal Care and Use Committee (approved number: CICUC2021-S0034). BMMs were isolated from the femurs and tibias of 6-week-old C57BL/6 male mice, as previously described [45]. Briefly, isolated cells were cultured in α-MEM (Welgene Inc., Gyeongsan, Republic of Korea) supplemented with 10% fetal bovine serum (FBS; Welgene Inc., Gyeongsan, Republic of Korea), 1% penicillin-streptomycin (Welgene Inc., Gyeongsan, Republic of Korea), and 10 ng/mL M-CSF (Sigma-Aldrich, St. Louis, MO, USA) for 16 h. Thereafter, collected nonadherent cells were cultured in 30 ng/mL M-CSF for 48 h. Finally, adherent cells were used as BMM in the present study. To osteoclast differentiation, BMMs were cultured at a density 1 × 10^4^ cells/mL for 72 h in the presence of 30 ng/mL M-CSF and 50 ng/mL RANKL (Sigma-Aldrich, St. Louis, MO, USA) with or without 10, 25, and 50 μg/mL OFLEE.

### 4.3. Cytotoxicity of OFLSS in BMMs

Both mouse embryonic fibroblast L929 cells used as normal cells to measure the cytotoxicity and BMMs were cultured at a density of 0.5 × 10^5^ cells/mL in a 96-well culture plate for 24 h. Thereafter, both cells were treated with 1, 10, 25, 50 and 100 μg/mL OFLEE for 24 h, followed by the addition of 20 μL of 3-(4,5-cimethylthiazol-2-yl)-2,5-diphenyl tetrazolium bromide (MTT) (5 mg/mL; Thermo Fisher Scientific Inc., Waltham, MA, USA). After additional cultivation for 4 h and removal of the supernatant, 400 μL of dimethyl sulfoxide (Sigma-Aldrich, St. Luis, MO, USA) was added to dissolve the MTT crystals. Absorbance was measured at 570 nm using a microplate spectrophotometer (BioTek, Winooski, VT, USA). The cell viability assay was repeated thrice independently. 

### 4.4. TRAP Staining and Activity Assay

TRAP staining and TRAP activity assay were performed using a TRAP and ALP double-stain kit (TaKaRa Bio Inc., Kyoto, Japan), according to manufacturer’s instruction. Stained cells were photographed under a microscope (Leica Microsystems, Mannheim, Germany). Thereafter, TRAP-positive cells were counted at equal areas. In addition, the TRAP activity was assessed by a microplate spectrophotometer (BioTek Instruments, Winooski, VT, USA).

### 4.5. F-Actin Staining

BMMs were fixed with 4% paraformaldehyde solution (Sigma-Aldrich, St. Louis, MO, USA) for 15 min, washed with potassium phosphate-buffered saline (PBS, Welgene Inc., Gyeongsan, Republic of Korea) and treated with 0.3% Triton X-100 (Sigma-Aldrich, St. Louis, MO, USA). Thereafter, BMMs were stained with rhodamine-conjugated phalloidin (Thermo Fisher Scientific Inc., Waltham, MA, USA) for 1 h and then stained with 4′,6-Diamidino-2-phenylindole dihydrochloride (DAPI, Vector Laboratories, Inc, Newark, CA, USA) for 5 min. Images were taken by a confocal laser scanning microscope (LSM 800 with Airyscan; Carl Zeiss, Oberkochen, Germany).

### 4.6. Western Blotting

Total proteins were extracted from BMM under previously described culture conditions using a radioimmunoprecipitation assay buffer (RIPA) buffer (Cell Signaling Technology, Danvers, MA, USA) and quantified using bicinchoninic acid (BCA) protein assay (Thermo Fisher Scientific, Rockford, IL, USA). Equal amounts of each protein sample were loaded onto 10% sodium dodecyl sulfate-polyacrylamide gel and then were transferred onto polyvinylidene fluoride membranes (Millipore, Burlington, MA, USA) for performing the Western blot using specific antibodies. The following antibodies used in the present study were purchased from Santa Cruz Biotechnology (Dallas, TX, USA): MMP-2 (sc-13595), MMP-9 (sc-13520), total-NF-κB (sc-8008), TRAF6 (sc-8409), c-fos (sc-8047), NFATc1 (sc-7294), cathepsin K (sc-4835), IL-1β (sc-52012), IL-17 (sc-374218), RANKL (sc-377079), RNAK (sc-374360), and β-actin (sc-8432). The following antibodies used in the present study were purchased from Cell Signaling Technology (Danvers, MA, USA): phospho-NF-κB (#3036), COX-2 (#12282), iNOS (#13120S), TNFα (#3707), and IL-6 (#12912). The immunoreactive bands visualized by the ECL system (Thermo Fisher Scientific, Rockford, IL, USA) were imaged using MicorChemi 4.2 (Dong-Il Shimadzu Corp., Seoul, Korea).

### 4.7. Gelatin Zymography

An equal volume of conditioned medium acquired from BMMs under previously described culture conditions was resolved on zymogram gel composed of 10% polyacrylamide gel copolymerized with 0.2% porcine skin gelatin (Sigma-Aldrich, St. Louis, MO, USA). Zymogram gel was incubated in renaturing buffer at 37 °C for 72 h to induce the reactivation of MMPs, and was stained with Coomassie brilliant Blue R250 (Sigma-Aldrich, St. Louis, MO, USA). The gelatinolytic bands were captured using a digital camera.

### 4.8. ROS Detection

After incubation under previously described culture condition for BMMs, ROS were detected using 2′,7′-dichlorfluorescein-diacetate (H2DCF-DA; Sigma-Aldrich). Thereafter, images were taken under an inverted fluorescence microscope (Eclipse TE2000; Nikon Instruments, Melville, NY, USA).

### 4.9. Measurement of PGE_2_

After incubation under previously described culture condition for BMMs, PGE_2_ production was measured using a PGE_2_ Parameter Assay kit (R&D Systems Inc., Minneapolis, MN, USA), according to manufacturer’s instruction.

### 4.10. Animal Study

Female C57BL/6 mice (8-week-old) were housed in an experimental housing facility (23 ± 3 °C, relative humidity 55 ± 15%, and 12 h light dark cycle) and supplied with water and chow diet ad libitum. After adaptation for 1 week, animals were randomly divided into 5 groups (*n* = 5/group) as naïve, sham, OVX, OVX supplemented 5 mg/kg, and OVX supplemented 10 mg/kg. Under anesthesia using isoflurane (Baxter, Deerfield, IL, USA), the mice were surgically ovariectomized via dorsolateral incisions, as previously described [46]. In addition, sham was generated by bilateral dorsal laparotomy. One week later, experimental animals were supplied a vehicle (water 100 μL), 100 μL of 5 or 10 mg/kg OFLEE suspended in water by oral gavage once daily for 9 weeks. In addition, body weights of experimental animals were measured by every week for 9 weeks. At the end of the animal study, experiment animals were sacrificed by carbon dioxide inhalation. To verify the hepatotoxicity, ALT and AST were measured by mouse ALT ELISA kit (ab285263, Abcam, Cambridge, UK) and AST activity assay kit (ab105135, Abcam, Cambridge, UK), respectively, in serum collected from experimental animals, according to manufacturer’s instruction. In addition, the microarchitecture of the femur dissected from experimental animals was analyzed using the Quantume GX microCT imaging system (PerkinElmer, Inc., Hopkinton, MA, USA) in Gwangju branch of the Korea Basic Science Institute (Gwangju, Republic of Korea).

### 4.11. Histological Analysis

For decalcification of the femoral bones dissected from experimental animals, 15% ethylenediamine tetraacetic acid (EDTA) was used and replaced every week for 10 weeks. Then, femoral bones were rinsed with tap-water for 24 h and soaked in a series of 20, 50, 60, 70, 90 and 100% ethanol. Thereafter, decalcified femoral bones were embedded in paraffin and incised into 8 μm slices. The samples were stained with H&E and Masson trichrome.

### 4.12. Statistical Analysis

All data are presented as the mean ± standard deviation from at least three independent experiments. Differences between groups were examined for statistical significance using Student’s t-tests and one-way analysis of variance using SPSS software (version 27.0; SPSS, Inc., Chicago, IL, USA). Statistical significance was set at **p* < 0.05 and ***p* < 0.01.

## 5. Conclusions

In the present study, we demonstrated that OFLEE induced anti-osteoporotic effects through the inhibition of osteoclastic differentiation from BMM and the down-regulation of MMPs expression and activation that was mediated by the suppression of ROS production, pro-inflammatory cytokines, and the expression of RANK and RANKL. Furthermore, oral supplementation of OFLEE effectively suppressed bone loss without hepatotoxicity in osteoporotic animals generated by OVX. Taken together, these findings consistently suggest that OFLEE may be a promising candidate for the prevention of osteoporosis.

## Figures and Tables

**Figure 1 plants-12-00253-f001:**
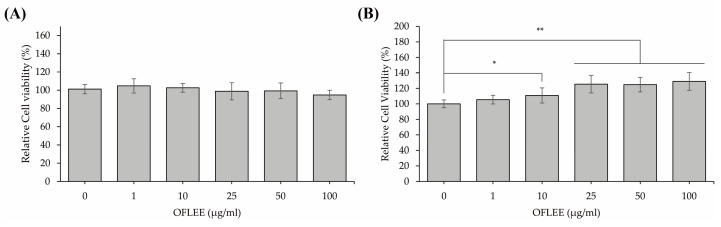
OFLEE does not induce cell death in either L929 fibroblast cells used as normal cells or BMMs. (**A**) OFLEE did not affect the viability of L929 fibroblast cells. (**B**) Quantities of 10–100 μg/mL OFLEE increase the viability of BMMs. Both mouse embryonic fibroblast L929 cells used as normal cells to measure the cytotoxicity and BMMs were cultured at a density of 0.5 × 10^5^ cells/mL in a 96-well culture plate for 24 h and were treated with 1, 10, 25, 50 and 100 μg/mL OFLEE for 24 h. Thereafter, MTT assay was performed to measure the cytotoxicity of OFLEE in L929 fibroblast cells and BMMs. Results are mean ± SD of three independent experiments (*n* = 3), * *p* < 0.05 and ** *p* < 0.01.

**Figure 2 plants-12-00253-f002:**
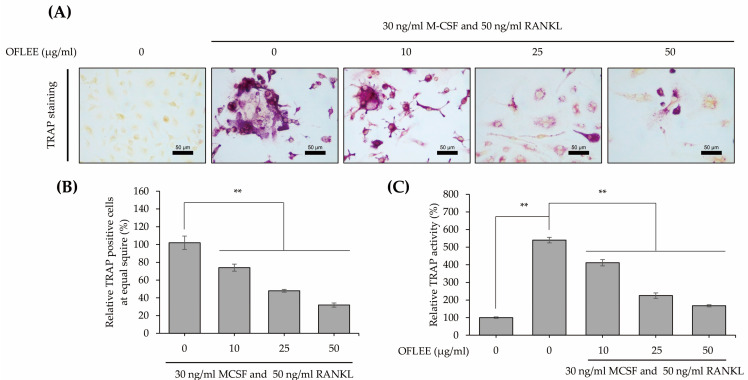
OFLEE suppresses TRAP-positive cells in the BMMs treated with M-CSF and RANKL. (**A**) OFLEE decreased the TRAP positive cells with multiple nuclei in the BMM treated with M-CSF and RANKL. (**B**) The number of TRAP positive cells were decreased by OFLEE treatment in the BMM treated with M-CSF and RANKL. (**C**) The OFLEE down-regulated TRAP activity in the BMM treated with M-CSF and RANKL. BMM was treated with 0, 10, 25, and 50 μg/mL OFLEE in the presence of 30 ng/mL M-CSF and 50 ng/mL RANKL for 72 h. Thereafter, the TRAP staining and TRAP activity assay were performed using a TRAP and ALP double-stain kit, according to manufacturer’s instruction. Stained cells were photographed under a microscope. Thereafter, TRAP-positive cells were counted at equal areas. In addition, the TRAP activity was assessed by a microplate spectrophotometer. Results are mean ± SD of three independent experiments (*n* = 3), ** *p* < 0.01.

**Figure 3 plants-12-00253-f003:**
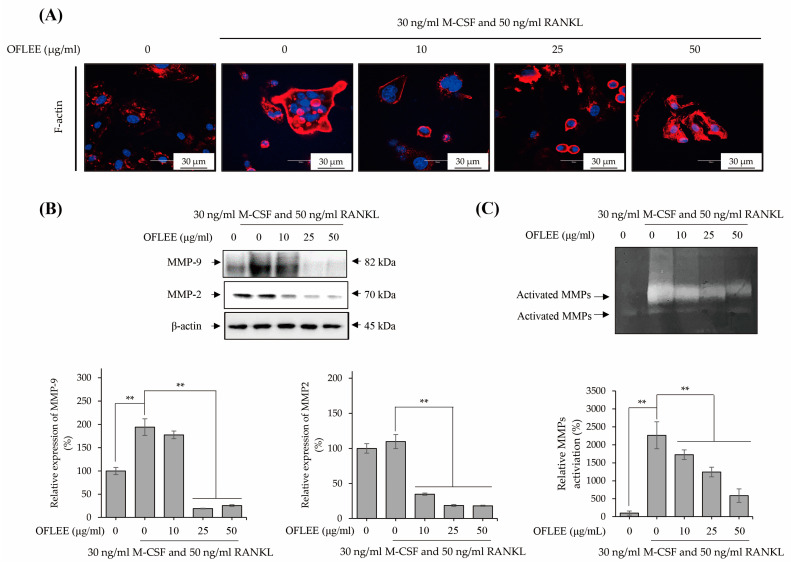
OFLEE suppresses the formation of the F-actin ring and the expression and activation of MMPs in the BMMs treated with M-CSF and RANKL. (**A**) OFLEE suppressed the formation of the F-actin ring in the BMMs treated with M-CSF and RANKL. BMMs were treated with 0, 10, 25, and 50 μg/mL OFLEE in the presence of 30 ng/mL M-CSF and 50 ng/mL RANKL for 72 h. Thereafter, BMMs were stained with rhodamine-conjugated phalloidin and then stained with DAPI. Images were taken by a confocal laser scanning microscope. (**B**,**C**) OFLEE inhibited the expression (**B**) and activation (**C**) of MMPs in the BMMs treated with M-CSF and RANKL. BMMs were treated with 0, 10, 25, and 50 μg/mL OFLEE in the presence of 30 ng/mL M-CSF and 50 ng/mL RANKL for 72 h. Thereafter, the expressions of MMPs in conditioned media were verified by Western blot using MMP-2 and MMP-9 antibodies. In addition, the activity of MMPs in conditioned media were verified by 0.2% porcine skin gelatin zymogram gel. The gelatinolytic bands were captured using a digital camera. Results are mean ± SD of three independent experiments (*n* = 3), ** *p* < 0.01.

**Figure 4 plants-12-00253-f004:**
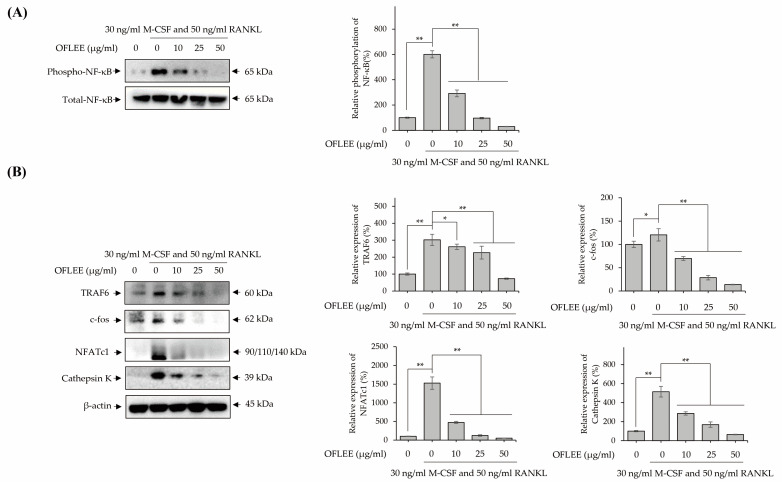
OFLEE suppresses the M-CSF and RANKL-induced osteoclastogenesis via the inhibition of TRAF6, NF-κB phosphorylation, c-fos, NFATc1, and cathepsin K in BMMs. BMMs were treated with 0, 10, 25, and 50 μg/mL OFLEE in the presence of 30 ng/mL M-CSF and 50 ng/mL RANKL for 72 h. Thereafter, Western blot was performed to verify the expression of TRAF6, NF-κB phosphorylation, c-fos, NFATc1, and cathepsin K. β-actin was used as internal control. (**A**) OFLEE suppressed the M-CSF and RANKL-induced NF-κB phosphorylation in BMMs. (**B**) OFLEE downregulated the osteoclastogenesis related transcriptional factors such as TRAF6, c-fos, NFATc1, and cathepsin K in the BMMs treated with M-CSF and RANKL. Results are mean ± SD of three independent experiments (*n* = 3), * *p* < 0.05 and ** *p* < 0.01.

**Figure 5 plants-12-00253-f005:**
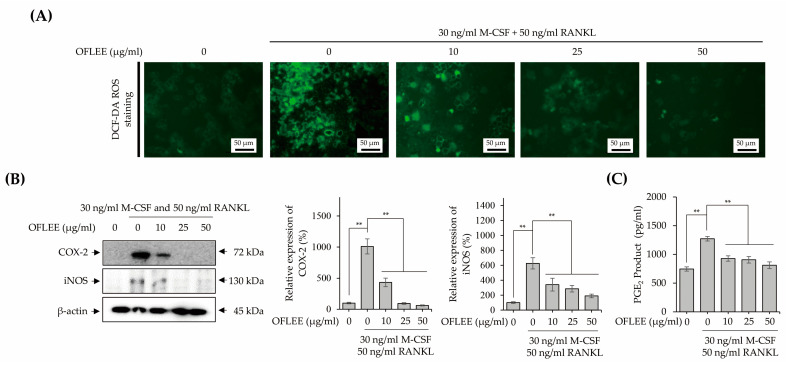
OFLEE suppresses the ROS production and the expression of inflammatory mediators such as iNOS, COX-2 and PGE_2_ in the BMMs treated with M-CSF and RANKL. (**A**) OFLEE suppressed the M-CSF and RANKL-induced ROS production in BMMs. BMMs were treated with 0, 10, 25, and 50 μg/mL OFLEE in the presence of 30 ng/mL M-CSF and 50 ng/mL RANKL for 72 h. Thereafter, ROS were detected using H2DCF-DA. Images were taken under an inverted fluorescence microscope. (**B**) The expression of inflammatory mediators such as iNOS and COX-2 was down-regulated by OFLEE in the BMMs treated with M-CSF and RANKL. (**C**) OFLEE decreased the production of PGE_2_ in the BMMs treated with M-CSF and RANKL. BMMs were treated with 0, 10, 25, and 50 μg/mL OFLEE in the presence of 30 ng/mL M-CSF and 50 ng/mL RANKL for 72 h. Thereafter, Western blot was performed to verify the expression of COX-2 and iNOS. β-actin was used as the internal control. In addition, PGE_2_ production was measured using a PGE_2_ Parameter Assay kit. Results are mean ± SD of three independent experiments (*n* = 3), ** *p* < 0.01.

**Figure 6 plants-12-00253-f006:**
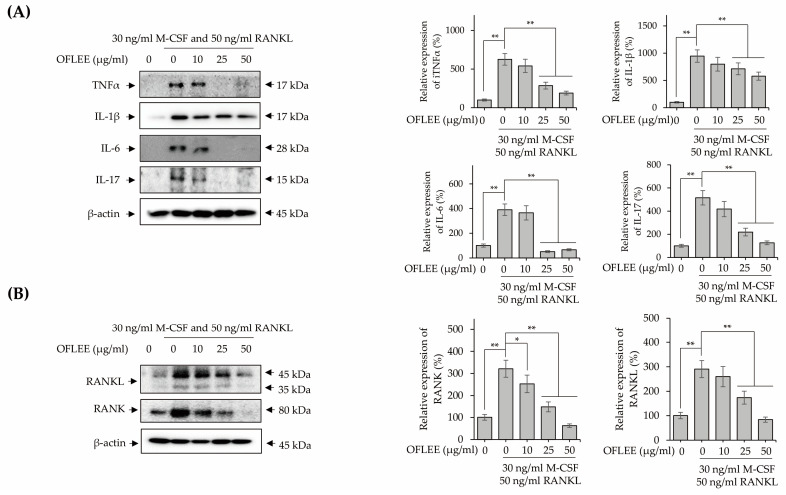
OFLEE down-regulated the expression of pro-inflammatory cytokines associated with the osteoclastogenesis in the BMMs treated with M-CSF and RANKL. BMMs were treated with 0, 10, 25, and 50 μg/mL OFLEE in the presence of 30 ng/mL M-CSF and 50 ng/mL RANKL for 72 h. Thereafter, Western blot was performed to verify the expression of TNFα, IL-1β, IL-6, IL-17, RANKL, and RANK. β-actin was used as an internal control. (**A**,**B**) OFLEE down-regulated the expression of pro-inflammatory cytokines including TNFα, IL-1β, IL-6, IL-17, RANKL, and RANK in the BMMs treated with M-CSF and RANKL. Results are mean ± SD of three independent experiments (*n* = 3), * *p* < 0.05 and ** *p* < 0.01.

**Figure 7 plants-12-00253-f007:**
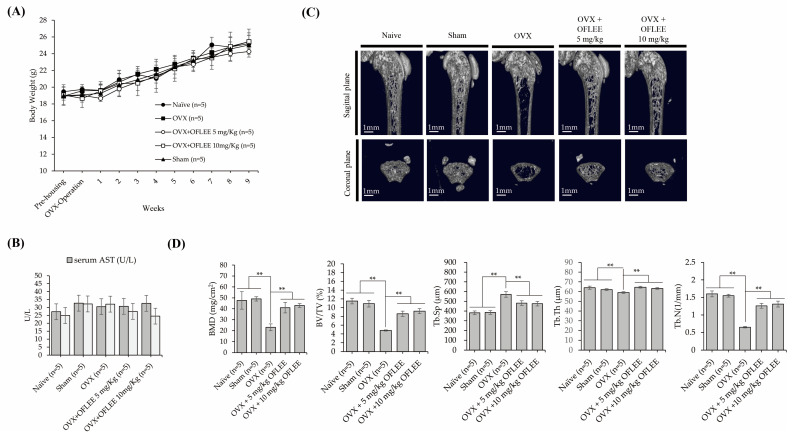
The oral supplementation of OFLEE prevented osteoclastic bone loss without hepatotoxicity in the experimental animals with OP generated by OVX. Animals (female C57BL/6 mice, 8-week-old) were randomly divided into 5 groups (*n* = 5/group) as naïve, sham, OVX, OVX supplemented 5 mg/kg, and OVX supplemented 10 mg/kg. In addition, animals were surgically ovariectomized to generate the animal model with OP. Thereafter, experimental animals were supplied a vehicle (water 100 μL), 100 μL of 5 or 10 mg/kg OFLEE suspended in water by oral gavage once daily for 9 weeks. Body weights of experimental animals were measured by every week for 9 weeks. To verify the hepatotoxicity, ALT and AST were measured by mouse ALT ELISA kit and AST activity assay kit, respectively, in serum collected from experimental animals. The microarchitecture of the femur dissected from experimental animals was analyzed using the Quantume GX microCT imaging system. (**A**) Body weight of experimental animals had no statistical significance between all of the animal groups, during the experiment periods. (**B**) the concentration of ALT and AST in the sera collected from experimental animals had no statistical significance between all of the animal groups. (**C**) the oral supplementation of 5 mg/kg (*n* = 5) or 10 mg/kg (*n* = 5) OFLEE significantly suppressed the bone loss in the femoral bone of animals with OVX. (**D**) Trabecular morphometric parameters, such as bone mineral density (BMD), bone volume per tissue volume (BV/TV), trabecular number (Tb.N) and trabecular thickness (Tb.Th), were significantly decreased in the OVX group compared with naive, shame, OVX orally supplemented with 5 mg/kg or 10 mg/kg OFLEE groups, whereas trabecular separation (Tb.Sp) was increased in the OVX groups, compared with naïve, sham, OVX orally supplemented with 5 mg/kg or 10 mg/kg OFLEE groups. Results are mean ± SD of three independent experiments (*n* = 3), ** *p* < 0.01.

**Figure 8 plants-12-00253-f008:**
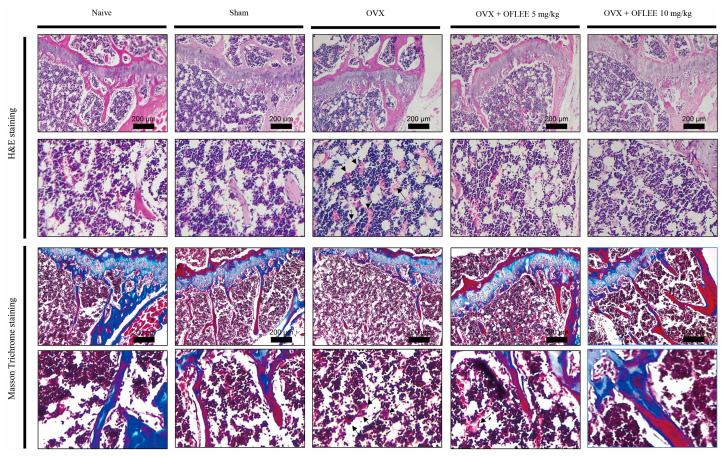
The oral supplementation of OFLEE prevented osteoclastic bone loss via the suppression of osteoclastogenesis in the experimental animals with OP generated by OVX. To perform the histological analysis, the femoral bones dissected from experimental animals were decalcified by 15% ethylenediamine tetraacetic acid (EDTA) for 10 weeks. Thereafter, decalcified femoral bones were embedded in paraffin and incised into 8 μm slices. The samples were stained with H&E and Masson trichrome. Hematoxylin and eosin staining showed the increase of empty lacunas and osteoclasts (indicated by black arrow) in the femoral bones dissected from the OVX group compared with naïve, sham, OVX orally supplemented with 5 mg/kg or 10 mg/kg OFLEE groups (**the upper panel**). Masson trichrome staining showed that new bone collagen fibers, stained a blue color, and mature bone, stained a red color, were higher than naïve, sham, OVX orally supplemented with 5 mg/kg or 10 mg/kg OFLEE groups compared with OVX group (**the lower panel**).

## Data Availability

The data presented in this study are available on request from the corresponding author.

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
