# Peer review of "The Ethanol Extracts of Osmanthus fragrans Leaves Ameliorate the Bone Loss via the Inhibition of Osteoclastogenesis in Osteoporosis"

_plants, 2023, doi:10.3390/plants12020253_

Round 1
Reviewer 1 Report
Please see the attachment.

Author Response
On behalf of authors listed in the manuscript entitled as Plants-2098968, I appreciate your encouragement for our study. Please find enclosed revised manuscript entitled as “The ethanol extracts of Osmanthus fragrans leaves ameliorate the bone loss via the inhibition of osteoclastogenesis in osteoporosis” by Yo-Seob Seo, HyangI Lim, Joeng-Yeon Seo, Kyeong-Rok Kang, Do Kyung Kim, Hyun-Hwa Lee, Deuk-Sil Oh, and Jae-Sung Kim for consideration of publication in “Plants”. We have appended a point-by-point response to each of reviewer’s comments.
Based on our comprehensive response to the comment of editor, we hope that our revised manuscript will be acceptable for publication in “Plants”.
Sincerely yours, on behalf of all co-authors,
Jae-Sung Kim Ph.D.
Institute of Dental Science,
Chosun University, Gwangju 61452, Republic of Korea
Tel: +82-62-230-7362; E-mail: js_kim@chosun.ac.kr

Reviewer 2 Report
In this manuscript, Seo and colleagues examine the in vitro role of OFLEE in the osteoclastic differentiation of BMMs, and they also provide in vivo evidence that OFLEE efficiently prevents osteoporosis in the OVX mouse model. OFLEE inhibits the differentiation of BMMs to osteoclasts by suppressing ROS production and pro-inflammatory cytokines. In addition, the authors confirm the anti-osteoporotic effects of OFLEE in the OVX-induced osteoporosis mouse model through daily OFLEE treatment for 9 weeks and suggest OFLEE may be a promising candidate to prevent osteoporosis. The major concern of this manuscript is that the conclusion needs further validation for the in vivo experiments.
1. The data presented in Figure 6 suggests BMD in the OFLEE treatment group is significantly reversed compared to the OVX group, the authors should also provide other micro-CT evaluation parameters, like BV/TV to confirm their data.
2. The authors have shown the effective function of OFLEE in preventing osteoporosis(in figure 6), and they hypothesize that OFLEE executes this function mainly through the suppression of osteoclastogenesis. However, the authors only show a significant bone loss rescue as an outcome, they need to perform trap staining or alternative experiments to check the osteoclastogenesis to confirm their mechanism.
3. A minor correction is that they should put scale bars on the representative images of micro-CT data(Figure 6C)
Author Response

(The authors gave the same response as above.)

Reviewer 3 Report
Reviewer comments of manuscript for second time, titled as “The Extracts of Osmanthus fragrans Leaves Ameliorate the Bone Loss via the Inhibition of Osteoclastogenesis in Osteoporosis”, which Submitted to plants
The authors should collect more information about usage of Osmanthus fragrans related to Bone Loss or others of your experiments.
Keywords should include anti-inflammatory with many cytokines and mediators
Why the authors extracted the materials using 80% EtOH?
Do you have a plan to find the active constituents from OFLEE?

Author Response

(The authors gave the same response as above.)

Round 2
Reviewer 2 Report
Dear authors,
My comments have been addressed and additional experiments have been completed.
I have a comment on this statement". We would like to perform the TRAP staining to verify the osteoclast in trabecular bone, but TRAP staining kit limited in the paraffin embedded bone tissues".
Instead of using TRAP staining kit, authors can apply the TRAP staining assays specific for paraffin sections to check osteoclasts in trabecular bones, and those protocols are available online.
As you made efforts with other histology experiments, I'm fine with your data and summary.